# Microstructures and Mechanical Properties of Laser-Sintered Commercially Pure Ti and Ti-6Al-4V Alloy for Dental Applications

**DOI:** 10.3390/ma13030609

**Published:** 2020-01-29

**Authors:** Yoshimitsu Okazaki, Akira Ishino

**Affiliations:** 1Department of Life Science and Biotechnology, National Institute of Advanced Industrial Science and Technology, 1-1 Higashi 1-chome, Tsukuba, Ibaraki 305-8566, Japan; 2IDS Co. Ltd., 3-5-4 Hongo, Bunkyo-ku, Tokyo 113-0033, Japan; a-isino@idscoltd.jp

**Keywords:** titanium materials, laser sintering, microstructure, tensile property, fatigue property, physical property, dental prostheses

## Abstract

To apply laser-sintered titanium (Ti) materials to dental prostheses with a three-dimensional structure such as partial dentures, we examined the microstructures and mechanical properties of commercially pure (CP) Ti grade (G) 2 annealed after laser sintering and laser-sintered (as-built) Ti-6Al-4V alloy. The tensile and fatigue properties of CP Ti G 2 annealed at 700 °C for 2 h after laser sintering were close to those of wrought CP Ti G 2 annealed at the same temperature after hot forging. The ultimate tensile strengths (σ_UTS_) of 90°- and 0°-direction-built CP Ti G 2 rods after laser sintering 10 times were 553 and 576 MPa and the total elongations (TE) of these rods were 26% and 28%, respectively. The fatigue strengths (σ_FS_) at 10^7^ cycles of the 90°- and 0°-direction-built CP Ti G 2 rods after laser sintering 10 times were ~320 and ~365 MPa, respectively. The ratio σ_FS_/σ_UTS_ was in the range of 0.5–0.7. The changes in the chemical composition and mechanical properties after laser sintering 10 times were negligible. The fatigue strength of the laser-sintered Ti-6Al-4V alloy was ~600 MPa, which was close to that of wrought Ti-6Al-4V alloy. These findings indicate that the laser-sintered CP Ti and Ti-6Al-4V alloy can also be applied in dental prostheses similarly to laser-sintered Co–Cr–Mo alloy. In particular, it was clarified that laser sintering using CP Ti G 4 powder is useful for dental prostheses.

## 1. Introduction

Metallic materials such as titanium (Ti) materials and cobalt–chromium–molybdenum (Co–Cr–Mo) alloys with excellent mechanical properties and structural stability have been widely used for dental prostheses (especially metal frames for partial dentures, complete dentures, and implant superstructures) [1,2,3]. Ti materials (commercially pure (CP) Ti and Ti alloys) with high biocompatibility have been widely used for orthopedic implants and dental prostheses. CP Ti for orthopedic implants is classified into grades (G) G 1 (O ≤ 0.18 mass %), G 2 (O ≤ 0.25 mass %), G 3 (O ≤ 0.35 mass %), and G 4 (O ≤ 0.4 mass %) in International Organization for Standardization (ISO) 5832-2 [4]. As the grade of CP Ti increases, the amount of trace elements such as oxygen (O) increases. 

Ti-6%Al-4%V alloy (here and hereafter values of alloy compositions indicate mass %) is also widely used in the medical field and is standardized in ISO 5832-3 [5]. Since the O concentration in laser-sintered Ti-6Al-4V alloy increases upon laser sintering [6], high-purity Ti-6Al-4V G 5 powder for additive manufacturing (AM) has been standardized in ASTM F 2924 [7]. The O concentration in this standard is defined according to the particle size of the Ti-6Al-4V alloy powder. In the Ti alloys containing Al, the O concentration is limited to ≤0.2% [5]. On the other hand, since CP Ti does not contain Al, it is expected that the O concentration can be increased to 0.4% as standardized in ISO 5832-2 [4]. Ti materials have the advantage of being lower in density and lighter than Co–Cr–Mo alloys. Dental prostheses (metal frames for partial dentures, complete dentures, and implant superstructures) fabricated using Ti materials are also suitable for patients sensitive to Co ions released from Co–Cr–Mo alloys [2]. 

The mechanical properties of dental-cast CP Ti and Ti alloys have been reported [8,9,10]. In conventional dental casts with Ti materials, a brittle (hard) surface oxide (alpha-case) layer and blow holes in the central part of casts are formed owing to the reaction of Ti materials with the mold material during the casting process. These casting defects cause problems such as the breakage of partial dentures and clasps. This hard alpha-case layer has also made it difficult to adapt Ti materials to crowns, bridges, and so forth [11,12]. In the case of using a computer-aided machining (CAM) milling system, it is difficult to manufacture complicated shapes, and long processing times are required [2]. Therefore, AM is expected to be a new technology for manufacturing Ti dental prostheses. In the AM of Ti materials, since no investment mold material is used, no reaction occurs between the mold material and the Ti material. Thus, as Ti has a strong affinity to O, for the AM of Ti materials, it is important to examine the change in the O concentration of Ti materials after laser sintering. However, there have been few studies on the mechanical and microstructural properties of laser-sintered CP Ti and Ti-6Al-4V alloys for dental prostheses. In particular, there are few reports on the fatigue properties of laser-sintered Ti materials. 

To obtain regulatory approval for dental prostheses and orthopedic devices produced by 3-D layer manufacturing in Japan, evaluation of the chemical composition, powder recycling, melting point, microstructure, tensile property, immersion property, and fatigue property is desirable [1]. We acquired experimental data according to these requirements for regulatory approval in dental prostheses.

The microstructures and mechanical properties of laser-sintered Co–Cr–Mo alloys for dental prostheses have been investigated [1,2,3]. These days, the tensile and fatigue properties of Co-25Cr-5Mo-5W (SP2) and W-free Co-28Cr-6Mo alloys are extremely superior to those of conventional dental-cast Co–Cr–Mo alloys [1]. The ultimate tensile strength (σ_UTS_) and total elongation (TE) of Co-28Cr-6Mo alloys are close to those of hot-forged Co-28Cr-6Mo alloys. The fatigue strengths (σ_FS_) at 10^7^ cycles of the 90°- and 0°-direction-built Co-28Cr-6Mo alloys are ~500 and ~600 MPa, respectively [1]. These superior mechanical properties result from the precipitation of fine intermetallic compound π-phase (lattice parameter a = b = c = 0.633 nm) particles in the grains and grain boundaries of the fine face-centered-cubic (fcc) matrix formed owing to rapid solidification [1]. The changes in the chemical composition and mechanical properties of Co-25Cr-5Mo-5W and Co-28Cr-6Mo alloys after laser sintering 20 times are also negligible. 

In this study, we investigated the chemical composition, immersion property, microstructure, tensile property, and fatigue strength of CP Ti G 2 annealed at 700 °C for 2 h after laser sintering for dental application with 3-D structures (metal frames for partial dentures, complete dentures, and implant superstructures). The purpose of this study was to compare the mechanical properties of laser-sintered Ti materials with those of laser-sintered Co–Cr–Mo alloy. To develop dental prostheses with high strength, these properties were also compared with those of laser-sintered (as-built) Ti-6Al-4V alloy. 

Moreover, to obtain similar two-phase equiaxed grains of the alpha (α)-phase [hexagonal-close-packed (hcp) structure] and beta (β)-phase [body-centered-cubic (bcc) structure] as the wrought Ti-6Al-4V alloy [13], the effect of the heat treatment temperature on as-built Ti-6Al-4V alloy rods was also investigated. 

In particular, we focused on the fatigue properties of laser-sintered and wrought Ti materials. The results obtained in this study are expected to be useful for the development of Ti dental prostheses by 3-D layer manufacturing.

## 2. Experimental Procedure 

### 2.1. Test Specimens

#### 2.1.1. Laser Sintering and Dental Casting

CP Ti G 2 (EOS (EOS GmbH Electro Optical System, Krailling, Germany) and TILOP (Osaka Titanium Technologies Co., Ltd., Osaka, Japan) powders were prepared by plasma and Ar gas atomization processes, respectively. The CP Ti G 2 powder was laser-sintered using a system comprising an EOS M290 or EOS M270 machine ((EOS GmbH Electro Optical System, Krailling, Germany), EOSPRINT v. 1.3 (EOS GmbH Electro Optical System, Krailling, Germany) and HCS v. 2.3.29 software (EOS GmbH Electro Optical System, Krailling, Germany), and the TiCP 30 μm FlexLine parameter set. Ti-6Al-4V (EOS) alloy powder and a CP Ti G 2 ingot for preparing conventional dental casts were used for comparison. The Ti-6Al-4V alloy powder was laser-sintered in an Ar atmosphere using a system comprising an EOS M290 machine, EOSPRINT v. 1.5 and HCS v. 2.4.14 software, and the Ti64 Performance M291 1.10 parameter set. 

The laser beam power (P) and the hatch spacing between scan passes (H) were 150–300 W (mainly 280–300 W) and 0.1–0.15 mm (mainly 0.13–0.14 mm), respectively. The laser scan speed (V) and powder stacking (deposited layer) thickness (T) were fixed from 1000 to 1300 mm/s (mainly 1200–1300 mm/s) and 0.02–0.04 mm (mainly 0.03 mm), respectively. The laser spot focus diameter was from 0.1 to 0.3 mm (mainly 0.1 mm). The volumetric energy density (E) = P/(H·T·V) was 50–120 J/mm^3^ (mainly 60 J/mm^3^). Cylindrical specimens built by laser sintering were cut from the support materials. 

Cylindrical specimens with a diameter of 9 mm and a height of 50 mm were fabricated by laser sintering under the above conditions using EOSINT M 290 and EOSINT M 270 machines on support materials using the CP Ti G 2 powders. As shown in Figure 1, to investigate the effect of the building direction of laser sintering, the building direction was set to 0° (hereafter, 0° direction), 45° (45° direction), and 90° (90° direction) for the base plate using the CP Ti G 2 powders [1]. CP Ti G 2 cylindrical specimens after laser sintering were heat-treated at 700 °C for 2 h followed by air cooling. This heat treatment (annealing) at 700 °C (±10 °C) for 2 h is recommended by the FlexLine EOS material data. These conditions are also widely used for wrought Ti materials [14]. In the actual laser sintering of dental prostheses, a certain percentage of virgin powder is added. The number of repeated uses of the same powder of up to 10 times in this work was decided considering the effect of a small amount of residual powder. In the case of laser sintering from two to nine times, two cylindical specimens with a diameter of 9 mm and a height of 50 mm were laser-sintered in the 90° direction in each sintering. 

For comparison, Ti-6Al-4V alloy was laser-sintered in the 90° direction with the EOSINT M 290 machine (as-built Ti-6Al-4V alloy). To obtain two-phase grains with the α (hcp) and β (bcc) phases, as seen in wrought Ti-6Al-4V alloy, the test specimens subjected to heat treatment were prepared from the laser-sintered Ti-6Al-4V alloy. The test specimens were heat-treated at 840, 860, 880, 900, 920, 940, 960, and 980 °C for 2 h followed by air cooling [14]. The microstructure after annealing was observed by optical microscopy and scanning electron microscopy (SEM, Talos Thermo Fisher Scientific, Tokyo, Japan, Quanta 200FEG; acceleration voltage, 15 kV). The phases were also identified by X-ray diffraction with Cu Kα radiation (Rigaku, SmartLab, Tokyo, Japan; tube voltage, 45 kV; tube current, 200 mA; 2°/min; scan range (2θ) of 30°–80°).

An Ar gas pressure dental casting machine manufactured by Wada Precision Dental Laboratories (EZ Titan, Osaka, Japan) and an alumina phosphate investment mold (SUPER-VEST-D, Okazaki Minerals and Refining Co. Ltd., Osaka, Japan) were used to manufacture the cylindrical dental casts. CP Ti G 2 ingots of 40 g (30 mm diameter and 13 mm height; GC Corporation, Tokyo, Japan) used for conventional dental casts were employed as reference materials.

#### 2.1.2. Laser-Sintered Powders and Ti Material Rods

Figure 2 shows (a) the particle size distribution and (b) SEM images of the CP Ti G 2 powders. Figure 2a also shows the D_10_, D_50_, and D_90_ particle sizes corresponding to 10%, 50%, and 90% of the cumulative distribution, respectively. The particle size distributions of the powders were measured using an LA-750 particle size analyzer (Horiba, Ltd., Tokyo, Japan) in accordance with ISO 133320 [15]. EOS CP Ti G 2 particles were larger than TILOP CP Ti G 2 particles. The particle size distributions of the virgin TILOP powders and 10-times-sintered CP Ti powders showed the same tendency. 

The changes in the D_10_, D_50_, and D_90_ particle sizes of Ti-6Al-4V powders when the number of repetitions in AM is incresaed from once to five times are small according to the literature [6].

The chemical compositions of these powders and laser-sintered (as-built) CP Ti G 2 and Ti-6Al-4V alloy, and dental-cast CP Ti G 2 rods are shown in Table 1. The H, N, O, Fe, C, Al, and V concentrations in the laser-sintered and dental-cast CP Ti G 2 rods and laser-sintered Ti-6Al-4V alloy rods were measured in accordance with Japanese Industrial Standard (JIS) H 1619 [16], JIS H 1612 [17], JIS H 1620 [18], JIS H 1614 [19], JIS H 1617 [20], JIS H 1622 [21], and JIS H 1624 [22], respectively. These chemical analyses were performed at Kobelco Research Institute, Inc. (Hyogo, Japan). Even when the number of repetitions in AM was increased from once to 10 times, the changes in metal concentrations (impurities) of the laser-sintered CP Ti G 2 rods were very small.

### 2.2. Evaluation of Physical Properties 

The liquidus temperature of the laser-sintered CP Ti was measured by differential thermal analysis (DTA, TG-DTA 2200SA, Bruker Corp., Kanagawa, Japan) [14]. Test specimens of 3 mm diameter and 1.5 mm height were cut from the laser-sintered CP Ti G 2. Heat flows in DTA were measured at a heating rate of 10 °C/min in Ar at a flow rate of 200 mL/min. The densities (ρ) of powder and laser-sintered CP Ti G 2 were measured using an Ultrapycnometer 1000 M-UPYC (Quantachrome Instruments, Kanagawa, Japan) by gas pycnometry in accordance with JIS R1620 [23]. The Vickers hardness (Hv) of each laser-sintered CP Ti G 2 rod at room temperature was measured with a Vickers hardness tester (FV-310, Future-Tech Corp., Kanagawa, Japan) at three to five points at a load of 10 kg.

### 2.3. Microstructural Observation 

The CP Ti G 2 annealed after laser sintering and laser-sintered (as-built) Ti-6Al-4V alloy were embedded in resin and polished to a mirrorlike finish with 200–4200 grit waterproof emery paper and an oxide polishing (OP-S) suspension. Then, each specimen was etched with nitric acid solution containing 3 vol% hydrogen fluoride. The microstructures of the CP Ti G 2 annealed at 700 °C for 2 h after laser sintering and laser-sintered (as-built) Ti-6Al-4V alloy were analyzed by optical microscopy (Nikon ECLIPSE LV150, Tokyo, Japan) and transmission electron spectroscopy (TEM, Hitachi HF-2000, Tokyo, Japan; acceleration voltage, 200 kV) with energy dispersive X-ray spectroscopy (EDS 2008 ver. 1.2 RevE, IXRF Systems, Tokyo, Japan). After etching, the surfaces of the annealed CP Ti G 2 and as-built Ti-6Al-4V alloy were observed by optical microscopy at magnifications of 50× and 400×. TEM was performed using disc-shaped specimens of 3 mm diameter, which were prepared by electrolytic polishing with 5 vol% perchloric acid +60 vol% methanol +35 vol% butanol solution under conditions of 30 V and 75 mA at −30 °C. After electrolytic polishing, the transverse cross-sectional structure was observed by TEM at magnifications of 15,000× and 60,000×. The fracture surfaces after the tensile and fatigue tests were observed by SEM.

### 2.4. Static Immersion Test

The specimens (n = 2), each with dimensions of 15 mm × 32 mm × 1 mm, were laser-sintered (as-built) and dental-cast CP Ti G 2 plates. Immersion tests were conducted at 37 ± 1 °C in an incubator using 0.1 mol/L lactic acid +0.1 mol/L NaCl solution (pH = 2.3) [1] in accordance with ISO 10271 [24] and JIS T 6115 [25]. The concentration of Ti released into the solution over 7 d was determined (ng/mL) by inductively coupled plasma mass spectrometry (ICP-MS, NexION 300D, PerkinElmer, Kanagawa, Japan; isotopic mass number of Ti, 48). An internal standard solution of Y (isotopic mass number, 89) was used for correction of the Ti concentration. The mean amount of Ti released (µg/cm^2^/week) and the standard deviation were calculated for two specimens.

### 2.5. Room-Temperature Tensile Tests

Uniform rod specimens (rod diameter, 3 mm; gauge length, 15 mm) were cut from cylindrical specimens with a diameter of 9 mm and a height of 50 mm [1,26,27]. Tensile test specimens for dental-cast CP Ti G 2 were also cut from cylindrical specimens without defects as determined by X-ray inspection. The tensile test specimens were pulled with an Instron 5567 testing machine at a crosshead speed of 0.5% of the gauge length (GL)/min until the proof stress reached 0.2%. The crosshead speed was then changed to 3 mm/min and maintained at this speed until the specimen fractured. The 0.2% proof stress (σ_0.2%PS_), σ_UTS_, total elongation (TE) at breaking, reduction in area (RA), and elastic modulus (E) were measured in tensile tests. The mean and standard deviation were calculated from the results of at least four specimens.

### 2.6. Fatigue Tests

Miniature hourglass-shaped rod specimens (3 mm minimum diameter and 50 mm total length) cut from cylindrical specimens were used for fatigue tests [1,28]. The fatigue tests were carried out with a sine wave at a stress ratio R(minimum cyclic stress (σ_min_)/(maximum cyclic stress (σ_max_)) of 0.1 and a frequency of 15 Hz in air. To obtain profiles of the relationship between σ_max_ and the number of cycles to failure N (S–N curves), the specimens were subjected to cycling at various constant maximum cyclic loads up to N = 10^7^ cycles, at which the specimens remained intact. The fatigue strength at 10^7^ cycles (fatigue limit, σ_FS_) was measured from the S–N curves. 

## 3. Results and Discussion

### 3.1. Chemical Compositions and Physical Properties 

Table 1 shows the chemical compositions of laser-sintered (as-built) CP Ti G 2 and Ti-6Al-4V alloy rods with virgin powders (hereafter, once-sintered CP Ti G 2 rod), and the compositions after laser sintering 10 times (10-times-sintered CP Ti G 2 rod) with the same CP Ti G 2 powders without the virgin powder added. The changes in the chemical composition of the 10-times-sintered CP Ti G 2 rod were negligible. The results shown in Table 1 were in good agreement with the FlexLine EOS material data sheet and the Ti-6Al-4V material mill test certificate. 

Figure 3 shows the oxygen (O) concentration in laser-sintered EOS CP Ti G 2 rods, laser-sintered TILOP CP Ti G 2 rods, and TILOP CP Ti G 2 powders as a function of the number of times of repeated laser sintering. The bars in Figure 3 indicate the standard deviation. In Figure 3, the results for the Ti-6Al-4V alloy were taken from the literature [6]. The increase in the O concentration of the laser-sintered CP Ti G 2 rods was found to be negligible. The O concentration in the laser-sintered Ti-6Al-4V alloy slightly increased from 0.11% in the virgin powder to 0.14% in the once-sintered alloy. On the other hand, in a previous report, the O concentration of the laser-sintered Ti-6Al-4V alloys was increased to up to 0.21% by laser sintering [6]. In particular, almost no increase in O concentration was observed after sintering 10 times. Thus, it was found that CP Ti powder can be recycled up to 10 times in the laser sintering of CP Ti. These tendencies were similar to those obtained with laser-sintered Co–Cr–Mo alloys [1].

The densities of EOS virgin powder and once-sintered and 10-times-sintered (as-built) CP Ti G 2 rods were 4.49, 4.50, and 4.50 g/cm^3^, respectively. The densities of TILOP virgin CP Ti G 2 powder and 10-times-sintered CP Ti G 2 rod were the same. These values are also consistent with the value (4.50 g/cm^3^) shown in the FlexLine EOS material data sheet. Moreover, these densities of laser-sintered CP Ti G 2 were close to that (4.5 g/cm^3^) of the wrought CP Ti G 2 ([29], p. 126). These values are much lower than the densities (8.4–8.8 g/cm^3^) of the laser-sintered Co–Cr–Mo alloys [1]. 

The liquidus temperatures of once-sintered and 10-times-sintered TILOP CP Ti G 2 rods measured by DTA were 1659 and 1662 °C, respectively. These liquidus temperatures were close to those (1663 ± 1 °C) of wrought CP Ti G 2 reported in the literature [14]. The liquidus temperature of CP Ti G 2 was much higher than those (1458–1463 °C) of laser-sintered Co–Cr–Mo alloys [1]. 

The amounts of Ti ions released from the once-sintered TILOP and dental-cast CP Ti G 2 plates were 1.90 ± 0.10 and 1.10 ± 0.05 μg/cm^2^/week, respectively. These values were close to the amount (1.2 μg/cm^2^/week) of Ti ions released from wrought Ti-6Al-4V alloy at pH = 2.3 reported in the literature [30]. 

The Hv values (n = 5) of transverse sections of the 10-times-sintered 90° EOS and once-sintered 90° TILOP CP Ti G 2 rods, and dental-cast CP Ti G 2 were 187 ± 4, 207 ± 6, and 169 ± 6, respectively. These values were close to the Hv (approximately 200–210) of dental-cast CP Ti G 2 in the literature [8,12].

### 3.2. Microstructure of Laser-Sintered Ti Materials

Figure 4 shows optical microscopy images of transverse (T) and longitudinal (L) sections of annealed CP Ti G 2 rods after laser sintering (90° and 0° directions). Similar microstructures were observed in the transverse and longitudinal sections of 0°- and 45°-direction-built specimens. As clearly shown in Figure 4e, the structure of CP Ti G 2 annealed at 700 °C for 2 h after laser sintering was similar to that of CP Ti G 2 annealed under the same conditions after hot forging. On the other hand, as shown in Figure 4f, the center part of the dental-cast CP Ti G 2 rod had an α phase with an hcp crystal structure [8]. Figure 5 shows TEM images of transverse sections of annealed CP Ti G 2 after 90°-direction-built laser sintering. The bcc structure (lattice parameters a = b = c = 0.331 nm) precipitated in the grain boundaries of the hcp structure (a = b = 0.295, c = 0.468 nm) matrix. The values of these lattice parameters were consistent with those in a handbook of Ti material properties ([29], p. 125). 

For comparison, optical microscopy and SEM images of laser-sintered Ti-6Al-4V alloy are shown in Figure 6a–d. The laser-sintered Ti-6Al-4V alloy had an acicular structure. Figure 6e,f show TEM images of transverse sections of 90°-direction-built laser-sintered Ti-6Al-4V alloy. TEM images of the laser-sintered (as-built) Ti-6Al-4V alloy show that it consisted of a fine martensitic (α’) needle-like microstructure (hcp, a = b = 0.295, c = 0.468 nm) formed owing to rapid solidification [31,32,33,34,35,36,37,38]. 

Figure 7 shows optical microscopy and SEM images of the Ti-6Al-4V alloy annealed at 840, 900, and 920 °C for 2 h after laser sintering of the alloy built in the 90° direction. In X-ray diffraction, most of the α-phase (hcp) contained a very small amount of the β-phase (bcc). Similar microstructures were observed in the optical images of specimens annealed at 860, 880, 940, 960, and 980 °C. The α-phase tended to become coarse with increasing annealing temperature. The annealed alloy retained its needle-like structure and did not change to a two-phase structure of α (hcp)–β (bcc), in contrast to the wrought Ti-6Al-4 V alloy [13,39]. Hv (n = 5) of each transverse section in the heat treatment range decreased from 342 to 316 and tended to decrease slowly with increasing annealing temperature.

### 3.3. Mechanical Properties of Laser-Sintered Ti Materials

Figure 8 shows the mechanical properties (σ_0.2%PS_, σ_UTS_, TE, and RA) of laser-sintered EOS CP Ti G 2 rods as a function of number of the repetitions of laser sintering. Hardly any effect of the number of repetitions on σ_0.2%PS_, σ_UTS_, TE, and RA was observed up to 10 repetitions. 

Table 2 summarizes the tensile properties (mean ± standard deviation) of the CP Ti G 2 annealed after laser sintering, the laser-sintered (as-built) Ti-6Al-4V alloy, and the dental-cast CP Ti G 2. The tensile properties of annealed (wrought) CP Ti G 2 and Ti-6Al-4V alloy after hot forging shown in Table 2 are taken from the literature [39]. The strengths of the 10-times-sintered CP Ti were similar to those of the once-sintered CP Ti. In particular, it was found that the tensile strengths of the annealed CP Ti G 2 after laser sintering were close to those of the annealed (wrought) CP Ti G 2 after hot forging. Table 2 also shows the tensile properties of wrought CP Ti G 2 specified in ISO 5832-3 [4]. The mechanical properties of the CP Ti G 2 annealed after laser sintering obtained in this study satisfied these standard values. However, σ_0.2%PS_ ≥ 500 MPa specified in the dental standard ISO 22674 was not satisfied [26]. These characteristics of the laser-sintered CP Ti G 4 materials (O ≤ 0.4%) with CP Ti G 4 powder were expected because the same annealed microstructure as the wrought CP Ti G 4 material was obtained. The elastic moduli of the laser-sintered and dental-cast CP Ti G 2 rods were 113 ± 2 and 115 ± 4 GPa, respectively. These values were lower than the value in ISO 22674 (≥150 GPa) [26].

Considering the dental application of high-strength Ti alloy, the mechanical properties of laser-sintered Ti-6Al-4V alloy are compared in Table 2. The tensile properties of the laser-sintered Ti-6Al-4V alloy were higher than those of the dental-cast Ti alloys [8,10]. The mechanical strength of laser-sintered Ti-6Al-4V alloy was also higher than that of wrought Ti-6Al-4V alloy [39]. On the other hand, the ductility of laser-sintered Ti-6Al-4V alloy was close to that of wrought Ti-6Al-4V alloy. The tensile properties of laser-sintered Ti-6Al-4V alloy fully satisfied the tensile properties (σ_0.2%PS_ ≥ 780, σ_UTS_ ≥ 860 MPa, TE ≥ 10%) specified in ISO 5832-3 [5]. The tensile properties obtained in this study were also similar to those reported in the literature [32,33,34,35,36,37]. The tensile strength of Ti-6Al-4V alloy annealed after laser sintering tended to be lower than that of the as-built Ti-6Al-4V alloy [36,38]. To increase the ductility of alloys, it is effective to improve the morphology of the needle-like microstructure by heat treatment [33,34,35].

The Ti-6Al-4V alloy has much lower fracture toughness (Charpy V-notch impact toughness) than CP Ti G 2 ([29], p. 238). The fracture toughness of laser-sintered Ti-6Al-4V alloy should be evaluated in the future.

Figure 9 shows SEM images of the fracture surfaces of laser-sintered CP Ti G 2 after the tensile test. Magnified images of the rectangular areas in Figure 9a,b are shown in Figure 9b,c, respectively. Dimples were observed on the fracture surfaces, as shown in Figure 9c. Similar fracture surfaces were observed on all the other laser-sintered CP Ti G 2 specimens.

### 3.4. Fatigue Strengths of Laser-Sintered Ti Materials

Figure 10a,b shows S–N curves of annealed CP Ti G 2 rods ((a) EOS and (b) TIROP powders) after laser sintering and dental-cast CP T G 2 (shown in Figure 10a). Figure 10b also shows S–N curves of CP Ti G 2 annealed after hot forging taken from the literature for comparison [39]. The fatigue strengths of the 90°-, 45°-, and 0°-direction-built CP Ti G 2 rods were ~330, ~290, and ~380 MPa, respectively. The effects of the laser sintering direction on σ_FS_ of CP Ti G 2 were considered to be small because the anisotropy factor of the fatigue strength was 0.87 ((fatigue strength in 90° direction)/(fatigue strength in 0° direction)). This value was similar to those obtained for the laser-sintered Co–Cr–Mo alloy (0.83–1.0) [1] and the anisotropy of σ_0.2%PS_ caused by forging of Ti-15Zr-4Nb-4Ta alloys (0.83–0.94) [40]. Thus, it was considered that the effect of the laser sintering direction is negligible for CP Ti G 2. The fatigue strength of the 90°-direction-built 10-times-sintered CP Ti G 2 was ~340 MPa.

Figure 11 shows S–N curves of laser-sintered (as-built) Ti-6Al-4V alloy and wrought Ti-6Al-4V alloy annealed at 700 °C for 2 h after hot forging; the curve for wrought Ti-6Al-4V alloy is taken from the literature for comparison [39]. The fatigue strength of the 90°-direction-built Ti-6Al-4V alloy was ~600 MPa, which is close to that of wrought Ti-6Al-4V alloy. The improvements in the fatigue strength and durability of dental prostheses (such as partial dentures and clasps) are effective in improving the needle-like structure and reducing the thermal stress caused by laser sintering. We would like to investigate in detail the other orthopedic applications of laser-sintered Ti-6Al-4V alloy in future studies.

Figure 12 shows SEM images of the surface of 0°-direction-built CP Ti G 2. A fatigue crack developed with a fatigue fracture from the periphery of the specimen, and river-like patterns and striations were observed, as shown in Figure 12e,f [38].

It was found that the strength and ductility of CP Ti G 2 annealed at 700 °C for 2 h after laser sintering were excellent owing to the dispersion of fine hcp and bcc particles. AM of CP Ti materials may be a promising new manufacturing technology to replace dental Ti casting as shown in Figure 13. The fatigue strength of the laser-sintered Ti-6Al-4V alloy was ~600 MPa, which is close to that of wrought Ti-6Al-4V alloy. Therefore, laser-sintered Ti-6Al-4V alloy can also be applied in dental prostheses such as high-durability partial dentures and clasps.

## 4. Conclusions

We evaluated the chemical composition, physical properties, microstructure, immersion property, tensile property, and fatigue strength of CP Ti G 2 annealed at 700 °C for 2 h after laser sintering and laser-sintered (as-built) Ti-6Al-4V alloy for dental applications. 

The changes in the chemical composition and mechanical properties of CP Ti G 2 after laser sintering up to 10 times were negligible. The rates of Ti ion release were low, with CP Ti and dental-cast CP Ti showing release rates of 1.90 ± 0.05 and 1.10 ± 0.05 μg/cm^2^/week, respectively. 

The fatigue strengths of the 90°-, 45°-, and 0°-direction-built CP Ti were ~330, ~290, and 380 MPa, respectively. The fatigue strength of the 90°-direction-built alloys after laser sintering 10 times was also ~340 MPa. The ratios of the fatigue strength at 10^7^ cycles to the ultimate tensile strength (σ_FS_/σ_UTS_) were 0.5–0.7. 

Precipitates identified to be bcc particles (a = b = c = 0.322 nm) were found in the grain boundaries of the hcp matrix. The structure of the annealed CP Ti G 2 after laser sintering was similar to that of the annealed CP Ti G 2 after hot forging. 

It was found that the strength and ductility of annealed CP Ti G 2 after laser sintering are excellent owing to the dispersion of the fine bcc particles. AM of CP Ti materials (G 2 and G 4) may be a promising new manufacturing technology to replace dental Ti casting. In particular, it was clarified that laser sintering with CP Ti G 4 powder (O ≤ 0.4 mass %) is useful for dental prostheses.

The fatigue strength of the laser-sintered Ti-6Al-4V alloy was ~600 MPa, which is close to that of wrought Ti-6Al-4V alloy. Therefore, laser-sintered Ti-6Al-4V alloy can also be applied in dental prostheses such as high-durability partial dentures and clasps, and orthopedic implants such as artificial hip-joint stems.

## Figures and Tables

**Figure 1 materials-13-00609-f001:**
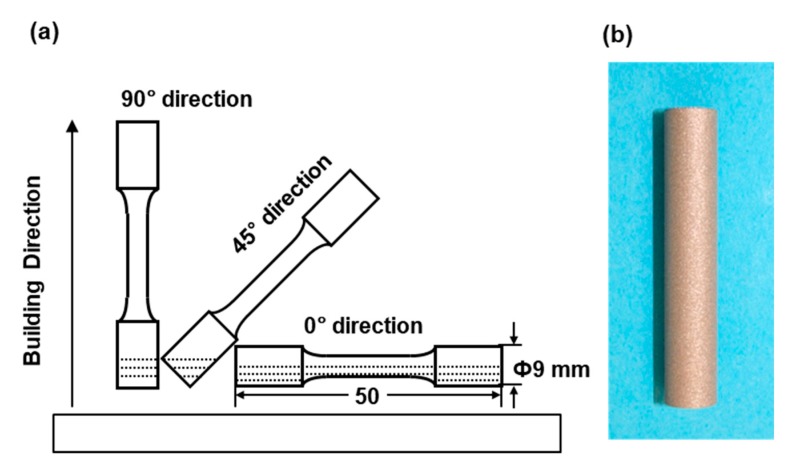
(**a**) Building directions of cylindrical specimens in laser sintering; (**b**) laser-sintered cylindrical specimens.

**Figure 2 materials-13-00609-f002:**
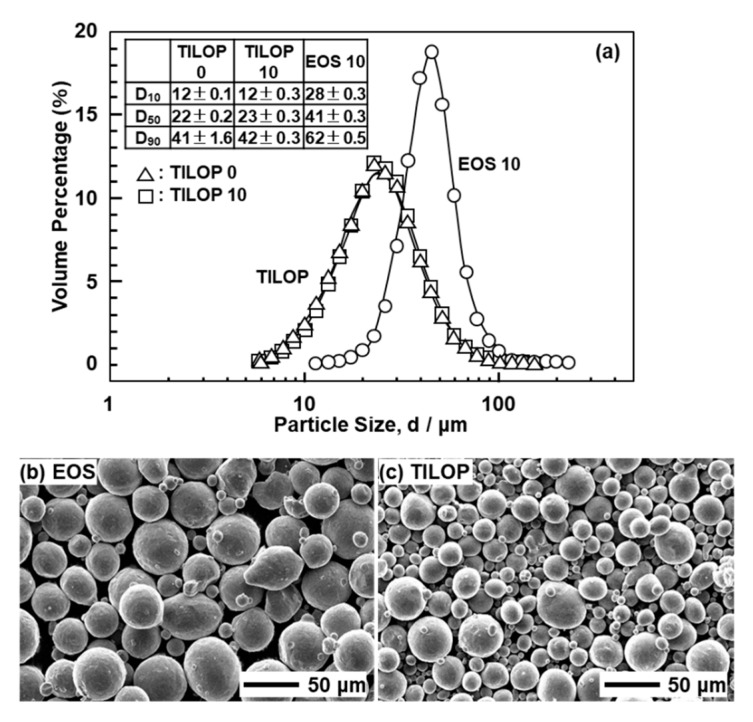
(**a**) Particle size distributions of 10-times-sintered EOS (EOS 10) and once-sintered (TILOP 0) and 10-times-sintered TILOP (TILOP 10) commercially pure (CP) Ti G 2 powders; SEM images of (**b**) 10-times-sintered EOS and (**c**) TILOP CP Ti G 2 powders.

**Figure 3 materials-13-00609-f003:**
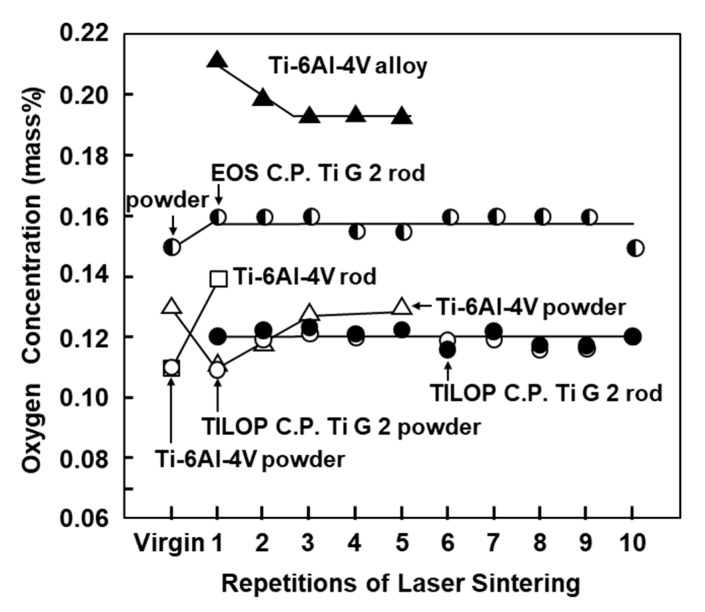
Effect of repeated laser sintering on oxygen concentration in Ti powders and laser-sintered Ti materials.

**Figure 4 materials-13-00609-f004:**
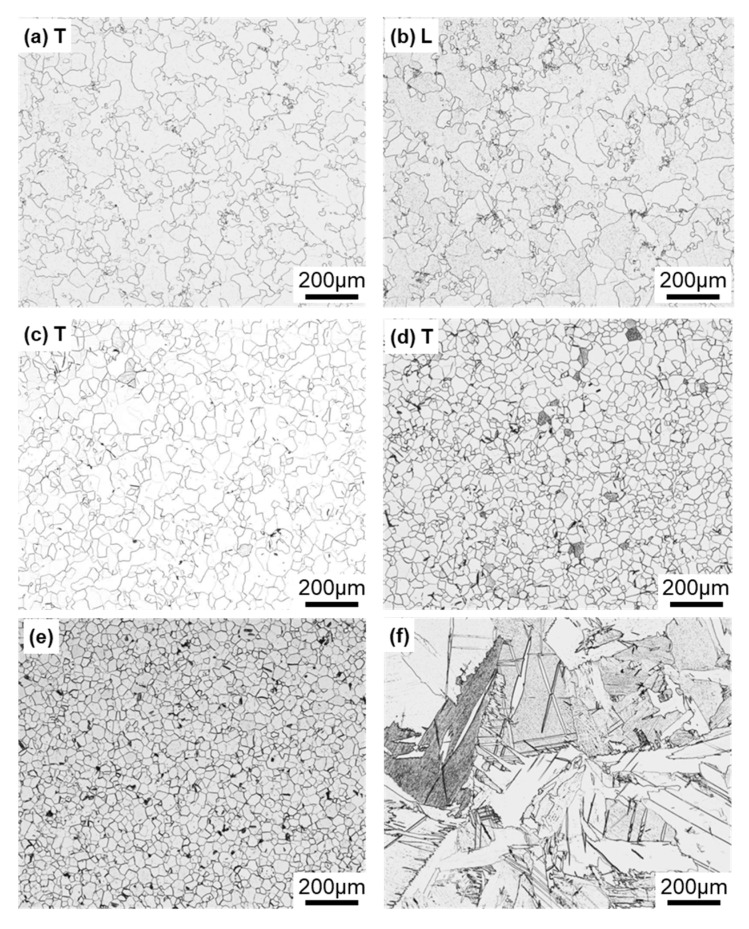
Optical microscopy images of (**a**,**b**) 10-times-laser-sintered EOS CP Ti G 2 and (**c**,**d**) once-sintered TILOP CP Ti G 2 built in (**a**–**c**) 90° and (**d**) 0° directions; (**a**,**c**,**d**) transverse (T) sections to the building direction and (**b**) longitudinal (L) section perpendicular to the building direction; (**e**,**f**) optical microscopy images of wrought and dental-cast (center part) CP Ti G 2 rods, respectively.

**Figure 5 materials-13-00609-f005:**
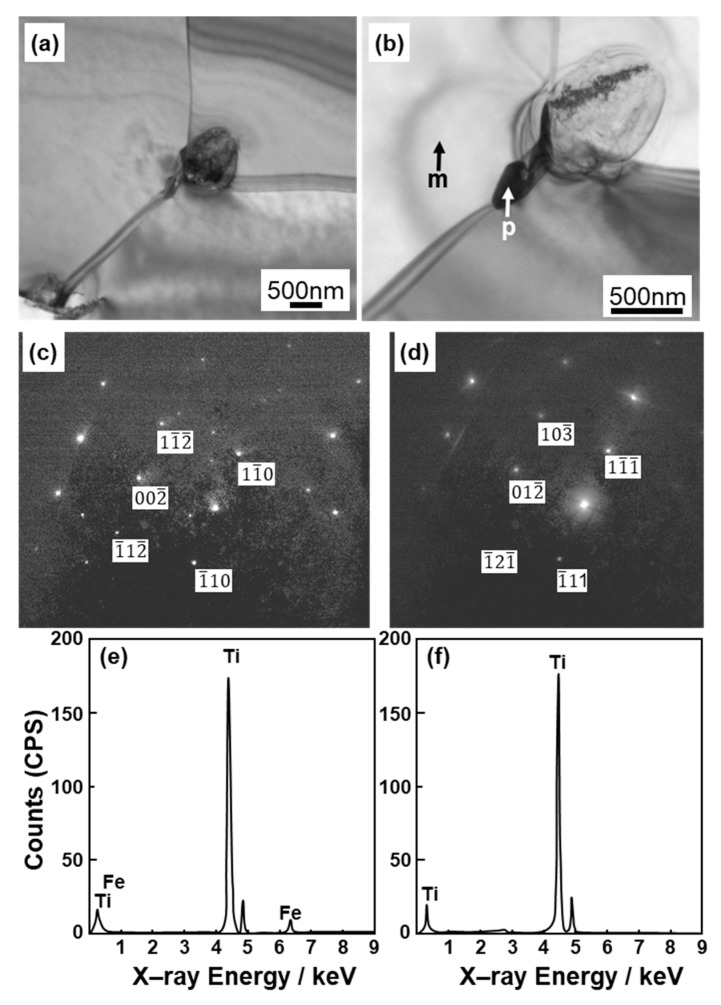
TEM images of transverse sections of (**a**,**b**) once-sintered TILOP CP Ti G 2 built in 90° direction; (**c**,**d**) electron beam diffraction patterns obtained at the location indicated by p (precipitation) and m (matrix) in (**b**), respectively; (**e**,**f**) EDS patterns of precipitate indicated by p and m in (**b**).

**Figure 6 materials-13-00609-f006:**
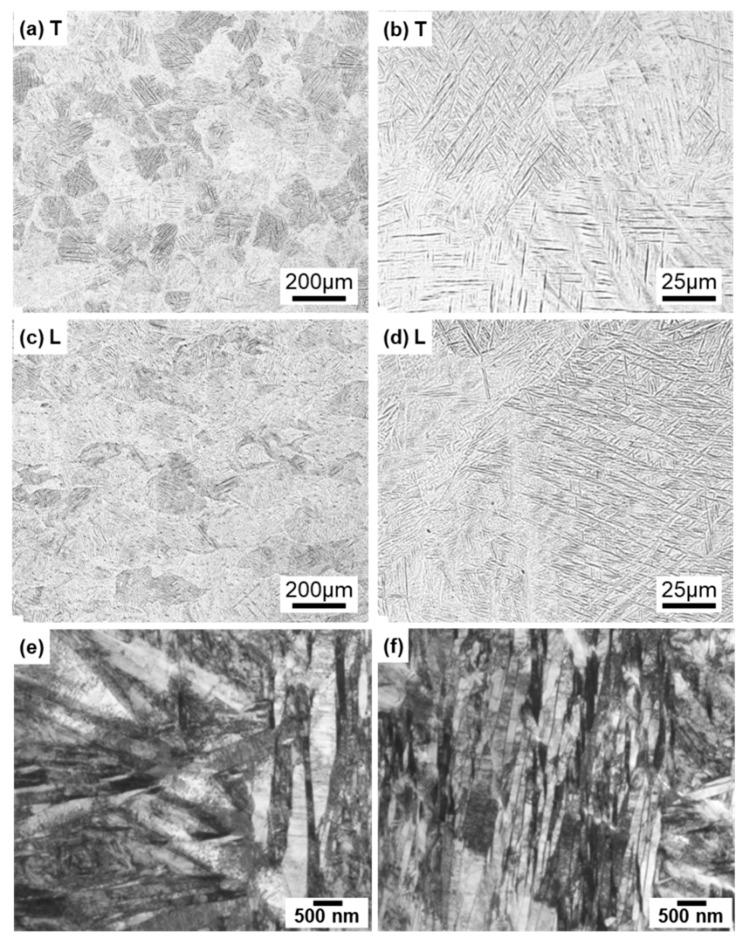
Optical microscopy images of once-sintered Ti-6Al-4V alloys built in 90° direction; (**a**,**b**) transverse (T) sections to the building direction and (**c**,**d**) longitudinal (L) sections perpendicular to the building direction; (**e**,**f**) TEM images of transverse section of once-sintered Ti-6Al-4V alloys built in 90° direction.

**Figure 7 materials-13-00609-f007:**
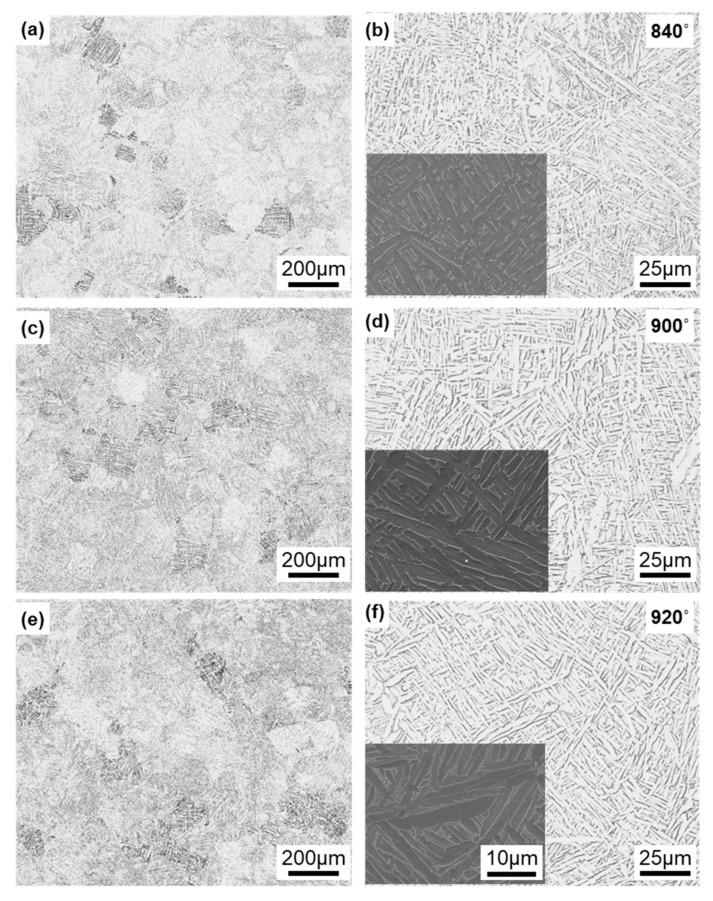
Optical microscopy and SEM images of Ti-6Al-4V alloy annealed at (**a**,**b**) 840, (**c**,**d**) 900, and (**e**,**f**) 920 °C for 2 h after laser sintering.

**Figure 8 materials-13-00609-f008:**
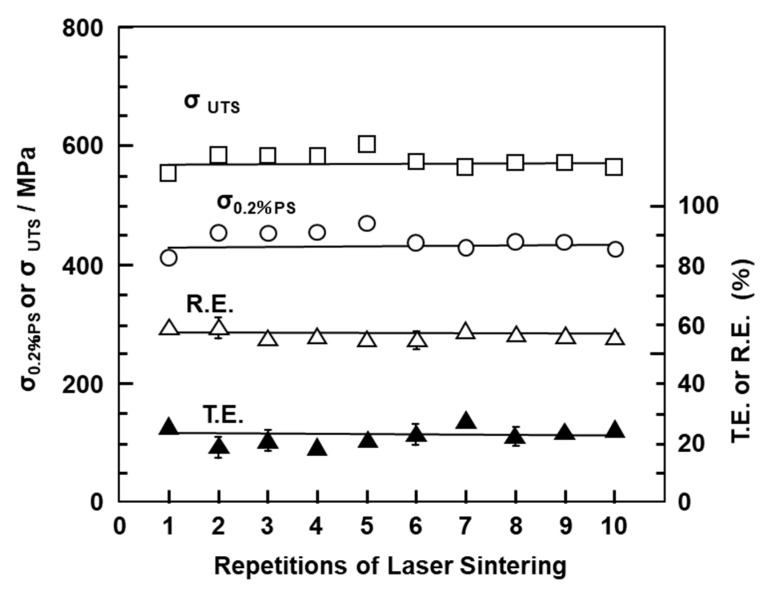
Effects of repeated laser sintering on mechanical properties (σ_0.2% PS_, σ _UTS_, TE, and RA) of laser-sintered EOS CP Ti G 2 rods.

**Figure 9 materials-13-00609-f009:**
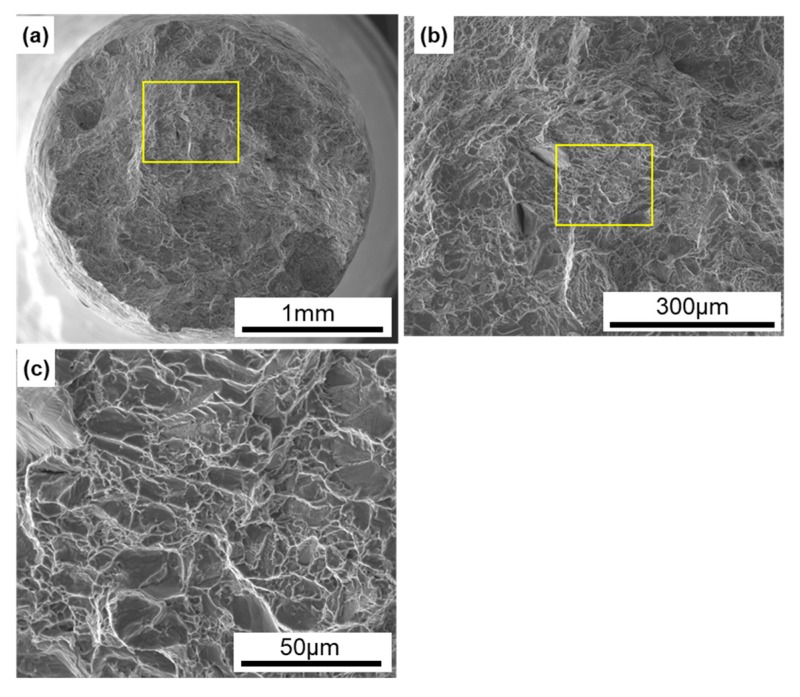
SEM images of fracture surfaces of the tensile-tested TILOP CP Ti G 2 (10-times-sintered); (**b**) magnification of rectangular area in (**a**), and (**c**) magnification of rectangular area in (**b**).

**Figure 10 materials-13-00609-f010:**
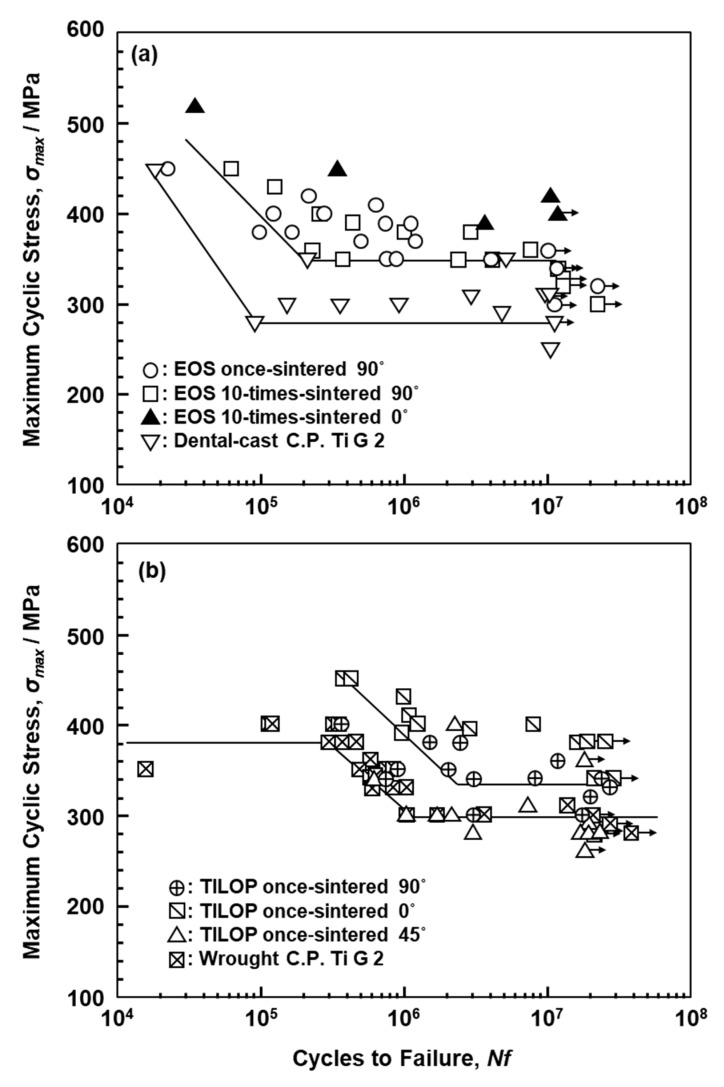
S–N curves of laser-sintered (**a**) EOS (once- and 10-times-sintered) and (**b**) TILOP (once-sintered) CP Ti G 2 rods; dental-cast CP Ti G 2 in (**a**) and wrought CP Ti G 2 in (**b**).

**Figure 11 materials-13-00609-f011:**
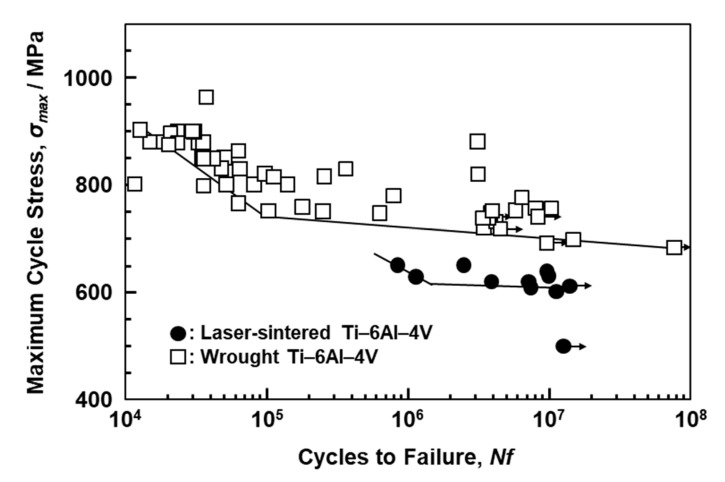
S–N curves of once-sintered Ti-6Al-4V alloy built in 90° direction and wrought Ti-6Al-4V alloy.

**Figure 12 materials-13-00609-f012:**
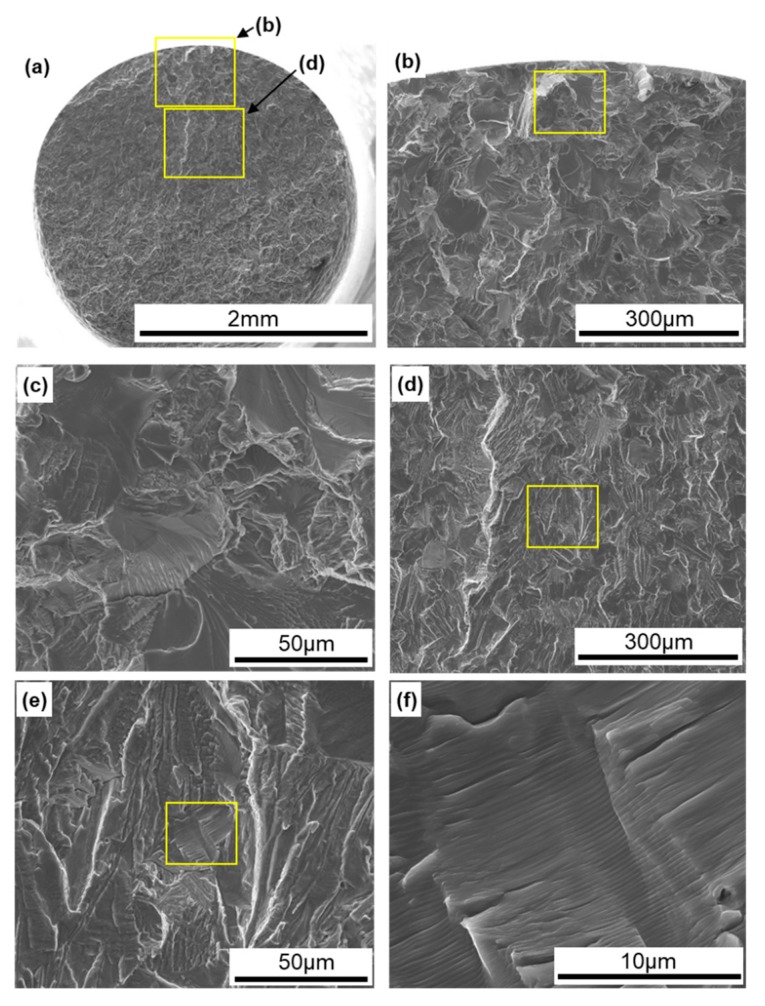
SEM images of fracture surfaces of fatigue-tested TILOP CP Ti G 2 (once-sintered 0°); (**b**) magnification of upper rectangular area in (**a**), (**c**) magnification of rectangular area in (**b**), (**d**) magnification of lower rectangular area in (**a**), (**e**) magnification of rectangular area in (**d**), and (**f**) magnification of rectangular area in (**e**).

**Figure 13 materials-13-00609-f013:**
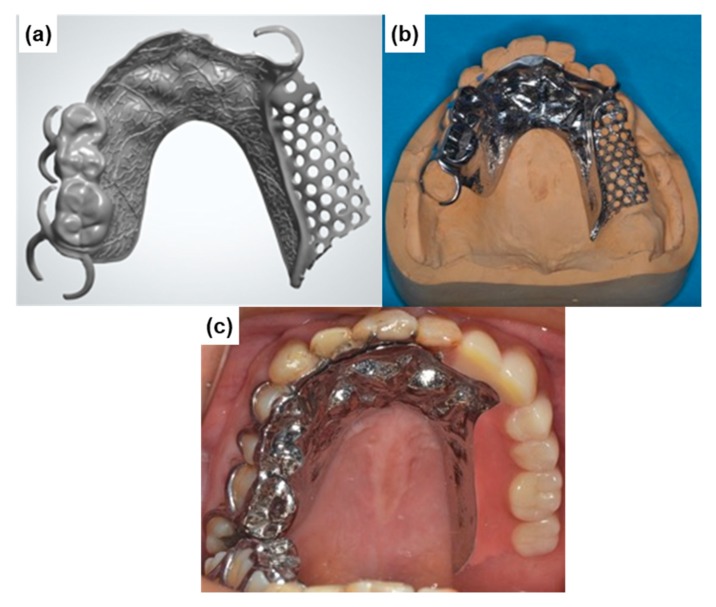
Partial dentures manufactured by laser sintering with CP Ti G 2 powder. (**a**) laser-sintered metal frame; (**b**) polished metal frame; (**c**) partial denture.

**Table 1 materials-13-00609-t001:** Chemical compositions (mass %) of CP Ti G 2 virgin powders, laser-sintered CP Ti G 2 and Ti-6Al-4V alloy, and dental-cast CP Ti G 2 rods.

Alloy	Al	V	Fe	O	N	H	C	Ti
EOS CP Ti G 2 virgin powder			0.16	0.15	0.01	0.002	0.01	Bal.
10-times-sintered powder			0.17	0.15	0.008	0.0013	0.009	Bal.
10-times-sintered CP Ti G 2 rod			0.17	0.16	0.017	0.0014	0.008	Bal.
TILOP CP Ti G 2 virgin powder			0.015	0.11	<0.005	0.0037	0.005	Bal.
10-times-sintered CP Ti powder			0.015	0.12	0.006	0.0035	0.005	Bal.
Once-sintered CP Ti G 2 rod			0.02	0.12	0.009	0.0032	0.002	Bal.
10-times-sintered CP Ti G 2 rod			0.02	0.12	0.007	0.0037	0.004	Bal.
Dental-cast CP Ti G 2 rod			0.103	0.17	0.007	0.0022	0.007	Bal.
Ti-6Al-4V powder	6.05	3.89	0.21	0.11	0.003	0.002	0.006	Bal.
Laser-sintered Ti-6Al-4V rod	5.93	3.91	0.20	0.14	0.022	0.002	0.007	Bal.

**Table 2 materials-13-00609-t002:** Tensile properties (σ_0.2%PS_, σ_UTS_, TE, and RA), fatigue strength (σ_FS_), and fatigue ratio (σ_FS_/σ_UTS_) of CP Ti G 2 and Ti-6Al-4V alloys manufactured under various conditions.

Specimen	σ_0.2%PS_/MPa	σ_UTS_/MPa	TE(%)	RA(%)	σ_FS_/MPa	σ_FS_/σ_UTS_
EOS CP Ti G 2						
Once-sintered 90°	412 ± 2	553 ± 3	26 ± 1	59 ± 1	320	0.58
10-times-sintered 0°	437 ± 2	576 ± 1	28 ± 2	55 ± 1	365	0.63
10-times-sintered 90°	426 ± 1	565 ± 1	25 ± 1	56 ± 1	320	0.57
TILOP CP Ti G 2						
Once-sintered 90°	445 ± 2	578 ± 3	27 ± 5	36 ± 10	330	0.57
Once-sintered 0°	432 ± 6	576 ± 5	26 ± 3	50 ± 1	380	0.66
Once-sintered 45°	419 ± 4	557 ± 3	22 ± 5	47 ± 6	290	0.52
10-times-sintered 90°	371 ± 4	481 ± 6	22 ± 8	41 ± 14	340	0.71
Dental-cast CP Ti G 2	351 ± 2.2	466 ± 3	30 ± 6	68 ± 5	290	0.62
Wrought CP Ti G 2	276 ± 6	410 ± 4	40 ± 2	60 ± 6	280	0.68
Laser-sintered Ti-6Al-4V	1171 ± 3	1305 ± 2	13 ± 1	31 ± 3	600	0.46
Wrought Ti-6Al-4V	849 ± 1	934 ± 1	16 ± 1	42 ± 3	680	0.73
ISO 5832-4	≥275	≥345	≥20

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
