# Peer review of "Microstructures and Mechanical Properties of Laser-Sintered Commercially Pure Ti and Ti-6Al-4V Alloy for Dental Applications"

_materials, 2020, doi:10.3390/ma13030609_

Round 1

Reviewer 1 Report

More detail about the actual application is not clear until the end of the paper when the applications for the Ti/Ti alloy components is made clear.  If the authors would identify the application from the start, the reader would be more interested in the course of the analysis and the results.  This is a minor change that would have a lot of impact on the quality of the paper. 

Also, by providing images of the laser-sintered AM components and the respective test specimens, visual impact will be added to the paper.  

Author Response

Reviewer 1

Thank you for the peer review of the manuscript, which has been revised in accordance with your comments. The corrected parts are shown in red in the manuscript. The modifications are as follows:

(1) We have revised the introduction to clarify the actual application and the purpose of the test.

   Figure 1, which shows the laser sintering direction of test specimens and laser-sintered specimens, and Fig. 13, which shows the image of an actual application (partial dentures), have been added.

(2) We have added a description and discussion of the results to make it easier for the readers to understand. The necessary references have also been added.

(3) Four figures (Figs. 1, 3, 8, and 13) have been added to make the manuscript easier to understand.

Reviewer 2 Report

General Comment. This reviewer prefers the use of “test specimen” or “specimen”, rather than “sample”, which has the traditional meaning in biostatistics of being a group of nominally identical, replicate, test specimens.

Introduction. (1) It might be mentioned that the Ti-6Al-4V alloy is sometimes referred to as the Grade 5 titanium alloy. (2) For prosthodontic dental applications (crowns and bridges), CP titanium and titanium alloys are cast, and the very hard alpha-case at the surface of the casting forms from reaction of titanium with the investment mold material and to a lesser extent with residual air in the titanium casting machine. The mechanical properties for dental-cast titanium and titanium alloys have been reported and should be included for comparison with the hot-forged and 3D-printed materials.

Experimental Procedure. (1) State the method used to obtain the compositions in Table 1, and indicate which composition determinations were obtained in the laboratory of the investigators. (2) For the dental-cast rods, the casting machine and the investment mold material should be noted. (3) Are there references to support the details of the protocol used for the 3D printing? (4) Can a reference be provided to support the 700°C annealing temperature for the laser-built CP Ti specimens? Indicate concisely the purpose of this heat treatment. (5) Was any ion milling or subsequent step performed after the electrolytic polishing to prepare the TEM specimens? (6) Was the aqueous lactic acid-NaCl solution used for immersion testing recommended in an ISO or another dental/biomedical standard? (7) At line 140, does “T.E.” correspond to the permanent percent elongation in tension? The full term should be noted parenthetically.

Results. (1) At line 158, the accuracy of the oxygen concentration measurements should be noted for the values reported in Table 1. (2) At lines 176 and 177, it should be noted how these lattice parameter values compare to those for the pure powder standard. (3) Does Figure 2 (f) present the near-surface or bulk microstructure of the dental-cast specimen. This should be noted in the figure legend.

Discussion. (1) Briefly explain the rationale for the 840°, 900° and 920°C temperatures and 2 hr time for heat treatment of the laser-sintered Ti-6Al-4V specimens. (2) An approximate value of cooling rate for the laser-sintered test specimens should be provided, with brief materials science-oriented comments about how this very rapid cooling accounts for microstructures of the printed CP Ti and Ti-6Al-4V specimens. (3) There should be a brief summary of the limitations of this investigation, with suggestions for future research. What are the factors that the authors consider to be of concern for the adoption of 3D printing by laser sintering to replace dental titanium casting?

Author Response

Reviewer 2

Thank you for the peer review of the manuscript, which has been revised in accordance with your comments. The corrected parts are shown in red in the manuscript. The modifications are as follows:

(1) We have modified the description of the test samples to “test specimens”.

(2) Ti-6Al-4V Grade 5 has been modified with reference to the introduction.

(3) We have added the problem of alpha-case generation on a titanium surface to the introduction.

(4) Casting machines and mold materials have been added to the test method.

(5) We have added the specialized analysis organizations that performed the analysis to obtain the values in Table 1 and modified the explanation and discussion of Table 1 to make it easier for readers to understand.

(6) We have added the laser sintering conditions, software, and so forth, to the test method.

(7) We have cited the basis for the 700℃ annealing conditions.

(8) We have pointed out that electrolytic polishing was performed to enable TEM observation.

(9) The specifications of the immersion test solution have been cited.

(10) The text has been modified to show that T.E. is the elongation at breaking in the tensile test. Test standards are also quoted. A description of T.E. can now be found in the introduction.

(11) The analysis method is in accordance with the JIS standards, so its accuracy is considered high. We have added a comparison with analysis values given by EOS powder manufacturers. To make it easier for the readers to understand, we added the changes in oxygen concentration with up to 10 repetitions of laser sintering to the new Fig. 3.

(12) The structure of the casting has been modified to show that it is at the center (bulk structure), not near the surface.

(13) We have added a discussion of the optical microstructures of laser-sintered Ti-6Al-4V alloy annealed at 840, 900, and 920℃. Since the cooling rate was not measured, it has been added that specimens were air-cooled.

(14) We have added comments so that the new findings and expected future developments based on the present study can be understood by the readers. The necessary references have been added.

(15) Two figures (Figs. 3 and 8) have been added to show the changes in the oxygen concentration and changes in the mechanical properties of commercially pure titanium and Ti-6Al-4 V alloy with repeated laser sintering, from which the readers can understand the test results for the recycling of Ti powders.

(16) We believe that the higher strength and elongation at breaking, and the absence of mold material make Ti materials an alternative to dental casting materials. Another advantage is that the fatigue strength is close to that of forged materials. We have carefully revised the entire manuscript to make these points clear.

(17) The necessary references have been added. Four figures (Figs. 1, 3, 8, and 13) have been added to make the manuscript easier to understand.

Reviewer 3 Report

The presented paper seems to be more technological report rather than a scientific paper, since no discussion is made there - just pure observations.

Normally I would say that the paper needs the discussion...

However considering the fact that this report is first of all very interesting , second, covers very broad spectrum of tests and data is taken professionally and well described, I would strongly suggest publishing it as it is  - it might be very interesting to the society and knowing the number of parameters as well as tests - the potential discussion might be controversial anyways.

The only one thing I would suggest is to describe more precisely - what authors mean by "sintered 10 times". I Understand that the powder was reused after making 10 processes of SLM but this is not obvious. 

Author Response

Reviewer 3

Thank you for the peer review of our manuscript. We are pleased with the comments you made on the manuscript. The corrected parts are shown in red in the manuscript. The modifications are as follows:

(1) The manuscript has been revised and Figs. 3 and 8 have been added so that the readers can understand the results of 10 repetitions of laser sintering.

(2) Four figures (Figs. 1, 3, 8, and 13) have been added to make the manuscript easier to understand.

Reviewer 4 Report

In this paper the authors investigate the chemical composition, immersion property, microstructure, tensile property, and fatigue strength of C.P. Ti G 2 annealed at 700 °C for 2 h after laser sintering for dental application. The topic is interesting The paper is of suffeicient novelty.

However the author don't investigate some other important properties such as Hardness, Fracture toughness, elasticity modulus etc. Besides they should explain in detail the utility of the investigation for the development of Ti dental prostheses by 3–D layer manufacturing.

Last, UTS and FS are subscripts of σ.

Author Response

Reviewer 4

Thank you for the peer review of our manuscript. We are pleased with the comments you made on the manuscript. The corrected parts are shown in red in the manuscript. The modifications are as follows:

(1) We have added data of hardness and elastic modulus. Regarding fracture toughness, we have added some results given in the literature. It is also a topic for future research.

(2) We have carefully reviewed and revised the entire manuscript to make it easier for the readers to understand.

(3) Although σUTS and σ FS look large, UTS and FS have subscripts of σ. Four figures (Figs. 1, 3, 8, and 13) have been added to make the manuscript easier to understand.

Round 2

Reviewer 4 Report

The paper has been sufficiently revised